# Remission of type 2 diabetes and improved diastolic function by combining structured exercise with meal replacement and food reintroduction among young adults: the RESET for REMISSION randomised controlled trial protocol

Kaberi Dasgupta ![ORCID],[1] Normand Boulé,[2] Joseph Henson,[3] Stéphanie Chevalier ![ORCID],[4] Emma Redman,[5] Deborah Chan ![ORCID],[6] Matthew McCarthy,[3] Julia Champagne,[6] Frank Arsenyadis,[3] Jordan Rees,[2] Deborah Da Costa,[1] Edward Gregg ![ORCID],[7] Roseanne Yeung,[8] Michelle Hadjiconstantinou,[3] Abhishek Dattani ![ORCID],[9] Matthias G Friedrich,[10] Kamlesh Khunti,[11] Elham Rahme ![ORCID],[1] Isabel Fortier,[1] Carla M Prado,[12] Mark Sherman,[13] Richard B Thompson,[14] Melanie J Davies,[3] Gerry P McCann ![ORCID],[9] Thomas Yates[3]

KD and TY are the co-lead investigators and co-corresponding authors. They contributed equally to this work.

For numbered affiliations see end of article.

**Correspondence to**
Dr Kaberi Dasgupta;
kaberi.dasgupta@mcgill.ca and
Dr Thomas Yates;
ty20@leicester.ac.uk

## ABSTRACT

**Introduction** Type 2 diabetes mellitus (T2DM) onset before 40 years of age has a magnified lifetime risk of cardiovascular disease. Diastolic dysfunction is its earliest cardiac manifestation. Low energy diets incorporating meal replacement products can induce diabetes remission, but do not lead to improved diastolic function, unlike supervised exercise interventions. We are examining the impact of a combined low energy diet and supervised exercise intervention on T2DM remission, with peak early diastolic strain rate, a sensitive MRI-based measure, as a key secondary outcome.

**Methods and analysis** This prospective, randomised, two-arm, open-label, blinded-endpoint efficacy trial is being conducted in Montreal, Edmonton and Leicester. We are enrolling 100 persons 18–45 years of age within 6 years' T2DM diagnosis, not on insulin therapy, and with obesity. During the intensive phase (12 weeks), active intervention participants adopt an 800–900 kcal/day low energy diet combining meal replacement products with some food, and receive supervised exercise training (aerobic and resistance), three times weekly. The maintenance phase (12 weeks) focuses on sustaining any weight loss and exercise practices achieved during the intensive phase; products and exercise supervision are tapered but reinstituted, as applicable, with weight regain and/or exercise reduction. The control arm receives standard care. The primary outcome is T2DM remission, (haemoglobin A1c of less than 6.5% at 24 weeks, without use of glucose-lowering medications during maintenance). Analysis of remission will be by intention to treat with stratified Fisher's exact test statistics.

## STRENGTHS AND LIMITATIONS OF THIS STUDY

⇒ RESET will evaluate diabetes remission alongside MRI-assessed diastolic function, an early indicator of the adverse impacts of type 2 diabetes, following a low energy diet and supervised exercise.
⇒ RESET will not distinguish the impacts of the individual dietary and exercise components.
⇒ We are focusing on young adults with type 2 diabetes, a group at elevated risk for diabetes complications.
⇒ Recruitment will be challenging due to competing responsibilities and lower numbers of young persons with type 2 diabetes compared with older age groups.

**Ethics and dissemination** The trial is approved in Leicester (East Midlands – Nottingham Research Ethics Committee (21/EM/0026)), Montreal (McGill University Health Centre Research Ethics Board (RESET for remission/2021-7148)) and Edmonton (University of Alberta Health Research Ethics Board (Pro00101088)). Findings will be shared widely (publications, presentations, press releases, social media platforms) and will inform an effectiveness trial.

**Trial registration number** ISRCTN15487120.

## BACKGROUND

The global prevalence of diabetes over 9% and is expect to surpass 10% by 2030.[1] More than 90% is type 2 diabetes mellitus (T2DM).

Diabetes is diagnosed in over 8% of Canadians and 6% of persons in the UK. The Da Qing T2DM prevention trial's 30-year follow-up demonstrated each year without T2DM translates into fewer cardiovascular disease (CVD) events.[2] T2DM remission is emerging as a potential goal in early-onset disease, defined as an A1C below diagnostic thresholds without glucose-lowering medications.

Health behaviour-based remission trials to date have focused on meal-replacement facilitated lowering of energy intake combined with exercise counselling.[3–6] The Look AHEAD (Action for Health in Diabetes) trial implemented a 1200–1800 kcal diet and an exercise target of 175 min/week. Its primary aim was reducing T2DM-related complications; remission, a secondary outcome, was 11.5% at 1 year and 7.3% at 4 years[7]; increased fitness predicted T2DM remission.[7] The DiRECT trial (Diabetes REmission Clinical Trial)[8–10] focused on remission within 6 years of a T2DM diagnosis through an 800 kcal meal replacement diet for 3–5 months, followed by a food-based weight maintenance diet. Participants received a step counter and were advised to increase their steps. Remission was 46% remission at 1 year[9] and over 30% at 2 years.[10] The DIADEM-I trial (Diabetes Intervention Accentuating Diet and Enhancing Metabolism) also focused on remission, among adults within 3 years of diagnosis through a dietary strategy similar to DiRECT but with more regular exercise counselling, incorporating both time and step goals as well as a goal of resistance training twice per week. DIADEM-I achieved over 60% remission at 1 year.[11]

Our RESET for REMISSION trial combines a meal replacement-facilitated dietary intervention with a supervised exercise intervention. Such interventions in T2DM demonstrated greater glycaemic lowering than unsupervised, counselling-based approaches,[12] with A1C lowering independent of weight loss[13] and conferring direct cardiac benefits in T2DM.[14 15] Isolated diastolic dysfunction as the earliest manifestation[16] of the concentric remodelling that typifies diabetic heart disease.[17] In the DIASTOLIC trial[18] (Diabetes Interventional Assessment of Slimming or Training tO Lessen Inconspicuous Cardiovascular Dysfunction), our UK team members compared low energy meal replacement diets against thrice weekly supervised aerobic exercise training over 12 weeks among young adults with T2DM to examine impacts on cardiovascular function, with remission as a secondary outcome. Seventy percent of those in the diet arm achieved remission, without impact on diastolic function; the exercise arm had little impact on remission but improved peak early diastolic strain rate,[18] a sensitive MRI-based diastolic function measure. In RESET for REMISSION, we will combine the dietary and exercise interventions and assess impacts not only on remission but also on MRI-based measures of cardiac function, another novel aspect of our trial. Moreover, in contrast to DIASTOLIC, we will combine supervised aerobic and resistance training to optimise lean mass preservation[19] with weight loss, and the potential for greater reductions in A1C than those observed for supervised aerobic exercise alone.[19 20] We hypothesise that this combination of a low energy meal replacement based diet and supervision for both aerobic and resistance training will have synergistic impacts on remission, enhance heart health, preserve lean mass and improve fitness.[19–21]

Diastolic dysfunction is more prevalent in younger adults with T2DM compared with weight matched and lean controls.[22 23] Compared with later-onset T2DM[24] and even type 1 diabetes,[25] young-onset T2DM has a more aggressive phenotype.[26] Younger adults represent at least 16% of T2DM internationally.[27] LookAHEAD studied persons 45–76 years of age and the average age in DiRECT was 53 years. DIADEM-I in contrast targeted adults 18–50 years of age. RESET for REMISSION will focus on those 18–45 years of age in cities in Canada and the UK. Even temporary T2DM remission could lead to fewer diabetes-related complications during what are supposed to be the most productive years.

## Aims

Among adults 18–45 years of age with obesity, within 6 years of a T2DM diagnosis, and not on insulin therapy, we will quantify the remission efficacy of a 12-week low energy diet combined with supervised exercise, followed by a 12-week phase of weight and exercise maintenance. We define T2DM remission as a haemoglobin A1c (HbA1c) value less than 6.5% (48 mmol/mol) at 24 weeks, without use of antihyperglycaemic medications during the prior 12 weeks, concordant with the international consensus definition.[3–6]

Through comprehensive cardiovascular MR (CMR), we are also assessing circumferential peak early diastolic strain rate, a sensitive measure[22] with excellent test–retest reproducibility[28]; end-diastolic mass to volume ratio, a marker of concentric left ventricular remodelling[29]; and measures of aortic distensibility, a key determinant of concentric remodelling in T2DM.[29] We will also ascertain impacts on a range of other measures (table 1) of insulin resistance, fitness, adiposity, cardiorenal parameters, diet, physical activity, mood and quality of life. We will evaluate participant perspectives (online supplemental appendix 1). Our results will guide the design of a longer-term effectiveness trial.

## METHODS
### Design

Prospective, randomised, two-arm, open-label, blinded-endpoint (PROBE) efficacy trial.

### Setting

Leicester (Leicester Diabetes Centre), UK; Montreal (Research Institute of the McGill University Health Centre-Centre for Outcomes Research and Evaluation, Canadian coordinating site), Canada; Edmonton (Alberta Diabetes Institute—University of Alberta), Canada.

**Table 1** Outcomes

| | Week 12 | Week 24 |
|---|:---:|:---:|
| **Primary outcome** | | |
| Diabetes remission at 24 weeks | | X |
| **Key secondary outcomes** | | |
| Other remission, glycaemic and insulin resistance measures | | |
| Diabetes remission at 12 weeks | X | |
| Haemoglobin A1c, fasting glucose and insulin, Homeostatic Model Assessment for Insulin Resistance | X | X |
| Main CMR measures | | |
| Left ventricular peak early diastolic strain rate (circumferential and longitudinal, MRI) | | X |
| End-diastolic mass to volume ratio | | X |
| Main fitness measure | | |
| $VO_2$ peak | X | X |
| Main fat and lean mass measures | | |
| Total fat and lean soft tissue mass (DXA) | X | X |
| Weight and BMI | X | X |
| Cardiometabolic indicators | | |
| Hypertension remission, systolic and diastolic blood pressure, heart rate | X | X |
| Total cholesterol, HDL, LDL, triglycerides | X | X |
| **Other secondary outcomes** | | |
| Renal function measures | | |
| Creatinine and estimated glomerular filtration rate | X | X |
| Urine albumin to creatinine ratio | X | X |
| Hepatic function measures | | |
| Includes alanine aminotransferase (ALT) and bilirubin | X | X |
| Depression, anxiety and distress | | |
| Hospital Anxiety and Depression Scale and Diabetes Distress Scale | X | X |
| Indirect calorimetry | | |
| Resting metabolic rate | X | X |
| Additional cardiac and aortic MRI-based measures | | |
| Longitudinal and circumferential measures of systolic strain, end systolic volume, ejection fraction, mean T1 time | | X |
| Cross-sectional areas and distensibility of ascending and descending aortae | | X |
| Additional measures of muscle mass and adiposity | | |
| Neck, hip and waist circumference | X | X |
| Visceral adipose tissue, pancreatic and liver fat percentages, subcutaneous adipose tissue, muscle mass (MRI) | | X |
| Dietary variables | | |
| Total energy and macronutrient intake (protein, carbohydrates, lipids) | X | X |
| Selected carbohydrate types (total sugars, starch, fibre), selected lipid types (saturated, monounsaturated, polyunsaturated, cholesterol), alcohol | X | X |
| Accelerometer-based physical activity measures and sleep (daily average) | | |
| Steps, overall acceleration, and intensity gradient metric | X | X |
| Minutes for each of sedentary, light and moderate to vigorous physical activity | X | X |
| Sleep time, duration of night, sleep efficiency (sleep time/duration of night) | X | X |
| Other exercise stress test measures | | |
| $VCO_2$ peak, maximum gradient achieved | X | X |

Continued

**Table 1** Continued

| | Week 12 | Week 24 |
|---|---|---|
| Tertiary outcomes | | |
| Bone measures | | |
| Total bone mineral density and bone mineral content (DXA) | X | X |
| Physical function | | |
| Handgrip strength | X | X |
| Short Physical Performance Battery | X | X |
| Dyspnoea scale | X | X |
| Overall health state | | |
| EuroQuol group 5-Dimensional 5-Level Questionnaire | X | X |
| WHO Disability Assessment Schedule 2.0 | X | X |
| Process evaluation | | |
| Acceptability, feasibility, facilitators, barriers and adherence to the intervention will be determined through interviews | | X |

BMI, body mass index; CMR, cardiovascular MR; DXA, Dual-energy X-ray absorptiometry; HDL, High-density lipoprotein; LDL, Low-density lipoprotein.

## Trial time frame

We recruited the first participant on 24 September, 2021; we plan to complete recruitment by 30 June 2024 and interventions and final evaluations by 31 January, 2025.

## Eligibility criteria

Adults with T2DM 18–45 years of age who were diagnosed less than 6 years previously, are not on insulin therapy, and who have a HbA1c value between 6% and 10% if taking other glucose-lowering medication or between 6.5% and 10% if not taking any glucose-lowering medications. Eligibility criteria are detailed in table 2.

## Recruitment

Recruitment is primarily through clinics and family practices within the collaborative networks of participating centres and publicity through social media and other forms of media.

## Run-in period

Candidates wear an accelerometer for seven consecutive days (wrist) and complete a 4-day food diary (three week-days and one weekend day). Five days of accelerometer wear and the 4-day food diary are required to move beyond the run-in period.

## Measurements

In-person evaluations occur before randomisation, at intervention period midpoint (12 weeks±1 week; no MRI study at this time point), and at intervention period end (24 weeks±2 weeks) (figure 1, table 1).

## T2DM remission and related measures

The primary remission outcome requires a HbA1c below 6.5% at intervention period end, without glucose-lowering medication during the prior 12 weeks.[3–6] Venous blood is sampled for HbA1c (high-performance liquid chromatography) measurement. Other related outcomes include HbA1c as a continuous variable, Homeostatic Model Assessment for Insulin Resistance and midpoint remission.

## MRI-defined cardiac structure and function

Using the 3-Tesla platform[30] (figure 2), we acquire cardiac cine images with retrospective electrocardiographic gating and an 18-channel phased-array cardiac receiver coil covering the left ventricle from base to apex (8 mm slice thickness, 2 mm gap with temporal resolution <50 ms and reconstructed to 30 phases). Using cmr42 software (CircleCardiovascular Imaging, Canada), we assess end-diastolic and end-systolic volumes, ejection fraction, myocardial mass, peak diastolic filling rate, systolic global longitudinal and circumferential strain, (simplified) long axis strain, and circumferential and longitudinal peak early diastolic strain rates (figure 3).

We quantify aortic stiffness, at the pulmonary artery bifurcation with simultaneous blood pressure recording, using Java Image Manipulation (Xinapse Software, Essex, UK).[29] We characterise myocardial tissue using a modified look-locker inversion-recovery sequence for a mid-level native T1 map. High native T1 time, and is a surrogate marker of diffuse interstitial fibrosis.

## Adiposity and lean measurements

We measure weight (nearest 0.1 kg) and height, waist, hip and neck circumferences (nearest 0.1 cm) and use DXA (GE Lunar iDXA) for total fat and lean mass (table 1). We acquire MRI images for visceral, subcutaneous, hepatic and pancreatic adipose tissue using the chemical shift encoded (DIXON) approach. Taylor's twin cycle hypothesis[31] emphasises hepatic fat accumulation in T2DM development, leading to insulin resistance along

**Table 2** Inclusion and exclusion criteria

| Inclusion criteria | |
|---|---|
| Age | 18–45 years, inclusive |
| Type 2 diabetes | Physician diagnosis more than 3 months and less than 6 years previously |
| Haemoglobin A1c | 6.5%–10%, inclusive if not taking glucose-lowering medication; 6%–10% if taking glucose-lowering medication |
| Body mass index | ► 30 kg/m²* to 45 kg/m², inclusive if White or Indigenous†<br>► 27 kg/m²* to 45 kg/m², inclusive if other background, including mixed |
| Weight stability | Weight changes of less than 5 kg over the prior 6 months |
| Walking ability | Able to walk without assists and to participate in structured exercise training requiring the lower limbs |
| Capacity | ► Able to understand written and spoken English and/or French<br>► Able to provide informed consent |
| Willingness | ► Willing to be randomised and able to participate<br>► Willing to attend supervised exercise sessions, if so randomised<br>► Willing to adopt low energy diet, including abstinence from alcohol, if so randomised<br>► Willing to self-monitor glucose and blood pressure at the required frequency, if randomised to the low energy diet plus supervised exercise arm |
| **Exclusion criteria** | |
| Other diabetes types | ► Type 1 diabetes<br>► Gestational diabetes<br>► Monogenic diabetes |
| Poorly controlled blood pressure | Resting systolic blood pressure greater than 150 mm Hg or resting diastolic blood pressure greater than 90 mm Hg diastolic |
| Weight loss interventions | ► Currently participating in a weight reduction programme in addition to routine care.<br>► Previous bariatric surgery. |
| Medications | ► Insulin therapy*<br>► Use of licensed weight loss medications<br>► Significant changes in glucose lowering medications in the prior 3 months, as judged by study physicians<br>► Steroids by mouth or injection |
| Self-reported allergies to components of meal replacement products | Milk protein and/or other relevant allergies |
| Dietary practices | Dietary practices that prohibit the use of meal replacement products |
| Pregnancy and lactation | ► Pregnancy<br>► Lactation<br>► Planning to become pregnant in the next 8 months |
| Eating disorder | Self-reported or diagnosed |
| Substance abuse | Alcohol, drugs |
| Estimated glomerular filtration rate | Less than 60 mL/min per 1.73 m² |
| Retinopathy | Receiving or requiring active treatment for retinopathy |
| Clinically manifest vascular disease | ► Myocardial infarction<br>► Stroke<br>► Peripheral vascular disease |
| Other cardiac disease | ► Heart failure<br>► Atrial fibrillation<br>► Pacemaker<br>► Implantable cardioverter defibrillator |
| Other conditions that could impact weight and/or safety | Active malignancy or other chronic disease |
| Run-in phase | ► Failure to complete at least 5 or requested 7 days of accelerometer wear<br>► Failure to complete a food diary for 3 weekdays and 1 weekend day |

*An exception is made for women who are on insulin therapy in case of pregnancy occurrence because of insulin's established safety profile in pregnancy, rather than because of inability to control glycaemia on oral agents alone. If these women are willing and able to use a reliable form of contraception, they may be enrolled.
†Term for the original peoples of North America and their descendants; includes First Nations, Inuit and Métis peoples.

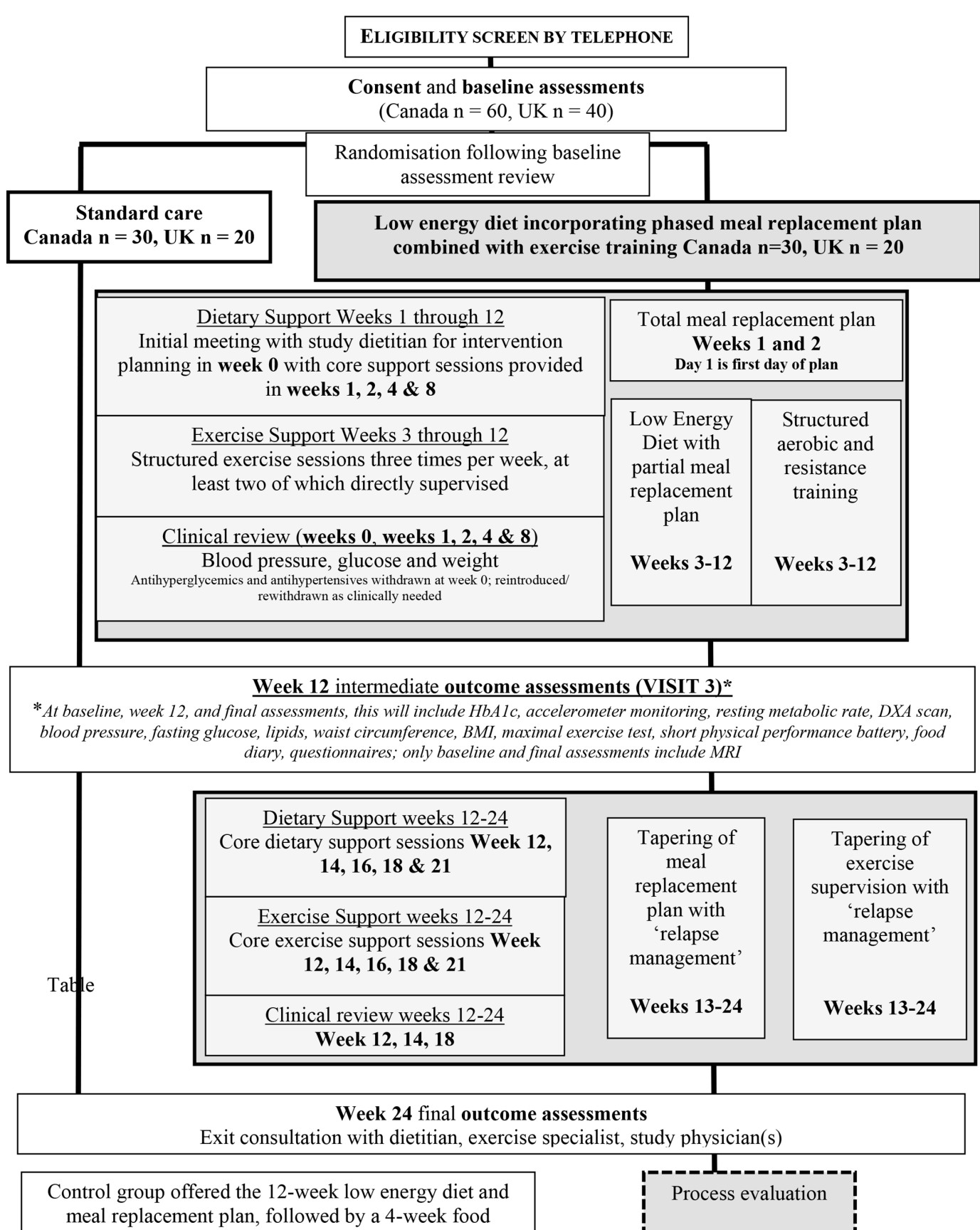

**Figure 1** Trial schematic. BMI, body mass index; HbA1c, haemoglobin A1c.

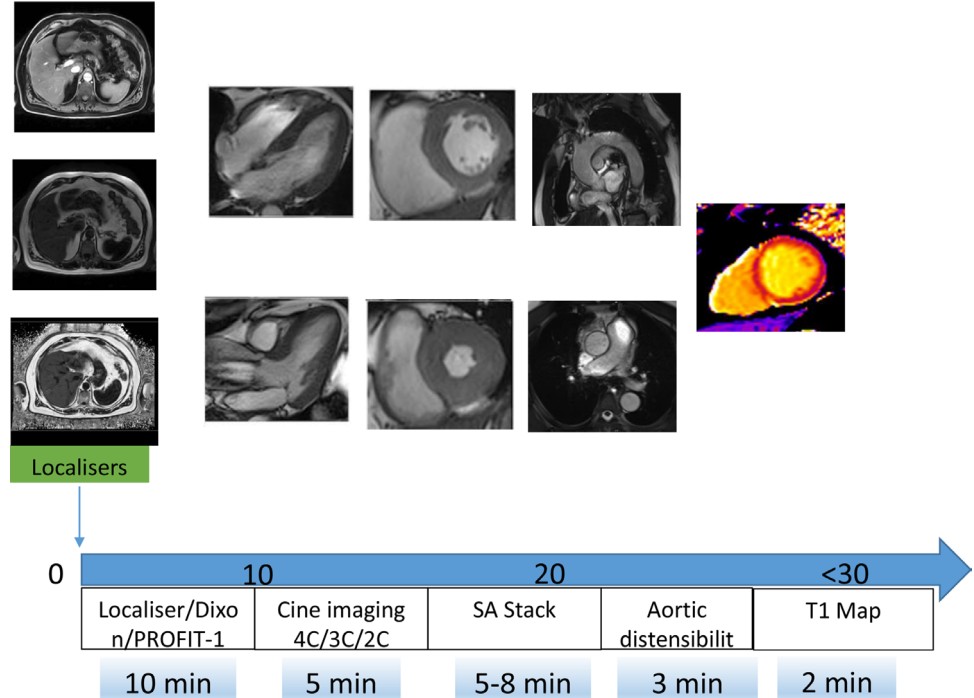

**Figure 2** Core MRI protocol of the RESET for REMISSION trial. The core protocol can be completed in under 30 min and comprises: localisers, fat-water imaging (Dixon), which allow assessment of subcutaneous and visceral adiposity (optional PROFIT -1 which adjusts liver fat measurement for fibrosis); cardiac cine imaging comprises long axes and complete coverage of the left ventricle and perpendicular to the thoracic aorta allowing calculation of myocardial volumes, mass, ejection fraction, strain/strain rates (see figure 3) and aortic distensibility; finally native T1 mapping in a single midventricular slice will be acquired as a surrogate marker of myocardial fibrosis. PROFIT1, simultaneous proton density fat fraction imaging and water T1 -mapping with low B1+ sensitivity; SA, short -axis.

with pancreatic fat accumulation that impairs beta-cell function.

### Optional MRI sequences

Time permitting, optional sequences include simultaneous PROton Density Fat Fraction Imaging and Water T1-Mapping with Low B1+Sensitivity of the liver (fibrosis), thigh (skeletal muscle volume) and heart (intramyocardial fat).[32] In Montreal, and time permitting at other sites, we perform oxygen sensitive CMR with hyperventilation and breath-holding, enabling ischaemia/microvascular dysfunction detection without contrast.[33] In Leicester, participants receive manganese contrast infusion, with repeated T1 mapping every 2.5 min for 30 min, to assess calcium handling.[34]

### Resting metabolic rate, fitness and strength

Following overnight fast, we conduct indirect calorimetry (ventilated hood system) for resting metabolic rate. We perform fixed speed treadmill exercise testing with increasing gradient (1% each minute) and a rolling average of 10 breaths for oxygen consumption and carbon dioxide production. The test continues to volitional exhaustion; 100% of age-predicted maximum heart rate (85% if using beta-blocker medication) plus respiratory exchange ratio ≥1.15; or ECG changes or symptoms of concern. We assess peak oxygen consumption, heart rate and blood pressure. The peak force output of handgrip

is determined across three measures in each hand. We apply the Short Physical Performance Battery,[35] WHO Disability Assessment Schedule 2.0 questionnaire[36] and the Medical Research Council dyspnoea scale (MRC Dyspnoea Scale).[37]

### Diet, physical activity and sleep

The 4-day food diary is repeated at intermediary and final evaluations for daily intakes (table 1). In addition to run-in, participants wear an accelerometer for the 7–14 days period proximate to the intermediary and final assessments. Accelerometer data is captured at 30 Hz (processed through the open source R program GGIR)[38] for daily averages of steps, overall acceleration, and intensity gradient metric, and total time at sedentary, light and moderate to vigorous intensities. Overnight wear permits capture of sleep time, night duration and their ratio (sleep efficiency).

### Cardiometabolic profile

We assess seated blood pressure (automated sphygmomanometer, averaged systolic and diastolic five sequential measures). We sample blood for lipid profiles, creatinine, electrolytes and liver function tests; and urine for the albumin to creatinine ratio. Among those without RAAS inhibition for albuminuria, we assess hypertension remission at intermediary and final assessments (blood

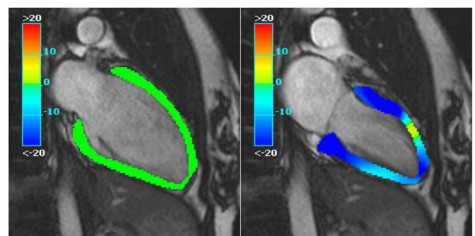
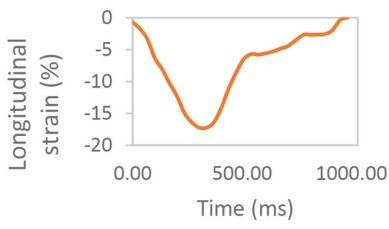
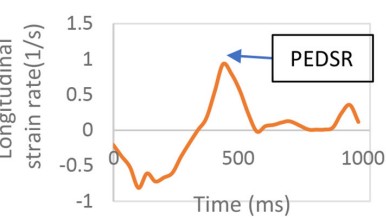
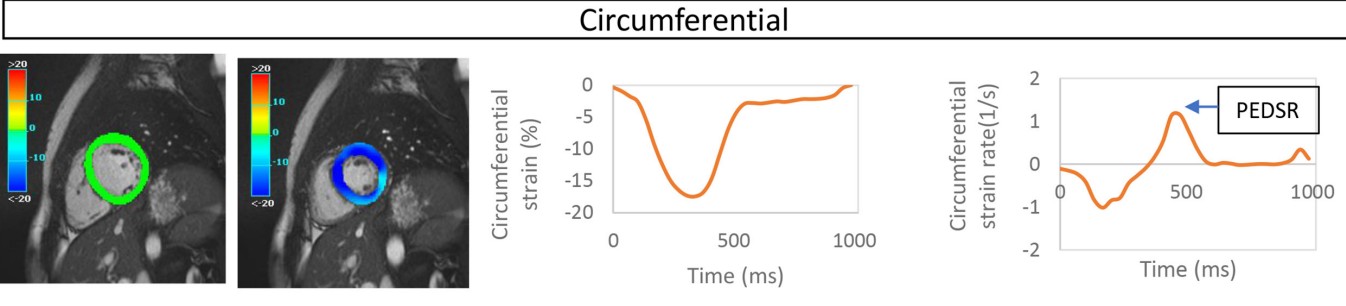

**Figure 3** A typical example of strain analysis from cardiac MRI cine imaging. Top panelbmjopen-2022-063888: Two-chamber view which allows derivation of longitudinal strain/strain rate; bottom panel mid ventricular short axis from which circumferential strain and strain rate is derived. Left: images showing color representations of longitudinal and circumferential strain at end-diastole and end-systole, with green showing 0%, light blue showing—10% and dark blue showing—20% strain (more negative indicating higher strain). Middle: graphs showing global longitudinal strain and global circumferential strain. Right: graphs showing longitudinal and circumferential strain rate throughout the cardiac cycle. PEDSR, peak early diastolic strain rate, a sensitive measure of diastolic relaxation and a key secondary outcome.

pressure values of 130/80 mm Hg without use of antihypertensive agents during prior 12 weeks.)

### Demographic factors, mental well-being and quality of life

Participants complete questionnaires on sex and gender, date of birth, ethnicity, employment, income, education, and marital status; anxiety and depression (Hospital Anxiety and Depression scale[39]); diabetes-related distress (Diabetes Distress Scale[40]) and quality of life (EuroQuol group 5-Dimensional 5-Level[41]).

### Intervention arm continuous blood glucose monitoring

To quantify the pattern of blood glucose responses to the intervention in detail, continuous glucose monitoring devices will be worn for 7–14 days preintervention, at intervention onset, midintervention and at the end of intervention, with computation of average glucose level, glucose variability (SD and coefficient of variation) and time within, above and below range.

### Data gathering for process evaluation

Through a qualitative descriptive approach, we will explore experiences of intervention arm participants.[42] We will use maximum variation sampling[43] to capture those who did and did not achieve remission and variations in age, sex, ethnicity and location. Our individual audiorecorded and transcribed in-depth interviews (online supplemental appendix 1) are informed by the normalisation process theory framework[44] and Capability, Opportunity, Motivation-Behaviour.[45] Sampling will continue until saturation of themes is achieved.[46]

### Randomisation and allocation concealment

We randomise participants individually, stratified by country and sex, in blocks of variable size. An independent statistician developed the randomisation sequences uploaded into the Research Electronic Data Capture (REDCap) system. A researcher blinded to the sequence unveils group allocation through an autogenerated randomisation button on REDCap.

### Experimental group
#### Full meal replacement diet for 2 weeks
##### Dietitian review

Participants meet with the dietitian to discuss the first 2 weeks of the intervention which include only Optifast products (Nestlé), totalling 800–900 kcal/day (30% protein, 50% carbohydrate and 20% fat). To minimise constipation, participants are recommended to drink at least 2 L of calorie-free fluid each day, and receive a fibre-based laxative, to use as needed. They are offered a digital

body weighing scale and asked to record weight weekly. They meet with the dietitian again in weeks 1 and 2 (virtually or in-person, as preferred).

### Physician review

The study physician meets with the intervention arm participant to discuss antihyperglycaemic and antihypertensive medication withdrawal,[47] given immediate lowering of glucose values and blood pressure with the dietary intervention. Glucagon-like peptide 1 receptors are stopped 1 week prior to the low energy diet. All other antihyperglycaemic medications are stopped the day that the diet is started. Except for ACE inhibitors or angiotensin receptor blockers prescribed for albuminuria (ie, RAAS blockade), participants stop antihypertensive medications on the day that the diet is initiated.

At weeks 1 and 2 follow-up (virtual or in-person), the physician reviews participant-recorded glucose and blood pressure data. Participants perform capillary blood glucose testing daily before breakfast and, on 1 day prior to physician follow-up, a 7-point glucose profile (measures before and 2 hours after food or meal replacement consumption and at bedtime). Antihyperglycaemic medications are not considered for reintroduction unless fasting morning capillary glucose is frequently 10 mmol/L or higher. Using the home blood pressure monitor provided, participants measure seated blood pressure each morning, after a 5 min rest period (second of two sequential measures). Antihypertensive agents are not be reintroduced unless systolic blood pressure values are generally 165 mm Hg or higher, as in the DiRECT trial.[48] RAAS blockade may be interrupted with symptomatic hypotension.

Participants who may become pregnant are required to use reliable contraception. This is discussed, with referral as needed.

### Partial meal replacement low energy diet and supervised exercise (weeks 3–12 inclusive)

#### Exercise physiologist

Starting at week 3, an exercise physiologist supervises 2 weekly sessions with aerobic and resistance components, at a designated facility. A third session is aerobic only, on site or elsewhere, as preferred, with self-monitoring (heart rate monitor and/or physical activity tracker provided). Participants increase exercise intensity and duration over time, with a goal of 60 min sessions.

Aerobic exercise on-site includes brisk treadmill walking (60%–80% of the maximum stress test heart rate). There is a high intensity interval walking option,[49] to maximise improvements in fitness, insulin sensitivity[50] and lean mass preservation.[51 52] With walking challenges (eg, plantar fasciitis), cycle ergometer may be offered.

Resistance training with machines or free weights includes one to three sets of four upper body exercises (bench press, seated row, shoulder press and pull down), three leg exercises (leg press, extension, flexion) and exercises targeting stability, function and posture, with

a goal of 8–12 repetitions per set. At 12 repetitions with good technique, the prescribed resistance is increased. Participants are instructed in resistance band use, in preparation for the maintenance phase.

#### Dietary component

Participants shift to partial meal replacement at week 3, with 800–900 kcal daily on non-exercise days and an additional 150–200 kcal from meal replacement products on exercise days. In addition to meal replacement products, the dietary 'real food' components are the equivalent of 125 mL of milk (or non-dairy alternative), careful attention to adequate protein intake from foods (meat, fish, egg or alternative), 1 portion of fruit and 2 portions of non-starchy vegetables each day. Because the variety of Optifast products are more limited in Canada than in the UK, ProtiDiet (soups, oatmeal and bars) are incorporated. Participants meet with the dietitian at weeks 4 and 8. The low energy diet is continued until week 12 and/or ideal weight (corresponding to body mass index of 25 kg/m$^2$ for white participants and those of Indigenous origin and 23 kg/m$^2$ in participants of non-white or mixed background.)

#### Physician review

Participants meet with the study physician at weeks 4 and 8. The systolic blood pressure threshold to consider antihypertensive reintroduction at this stage is 140 mm Hg, as in DiRECT.[48]

### Maintenance phase of intervention
#### Dietary component

By week 12's end, the dietitian and participant create an individualised plan for tapering meal replacement products, meeting again at weeks 14, 16, 18 and 21. The goal is to maintain any weight loss achieved in weeks 1–12. Meal replacement product tapering may be slowed, and product intake may even be increased again, in the event of weight regain.

#### Exercise component

Using monitoring tools, participants aim for 150 min of moderate to vigorous physical activity weekly, including three 30 min dedicated sessions at 60%–80% of the maximum heart rate . Participants may use the study exercise facility or engage in home-based or community-based exercise such as brisk walking and resistance training facilitated by bands. Participants have virtual or telephone contact with exercise physiologists at weeks 14, 16, 18 and 21. Those with less than 60 min of planned aerobic exercise and/or less than one resistance training session per week are asked to return for additional supervised sessions.

#### Physician monitoring

The study physician meets with participants at weeks 14 and 18. The thresholds for medication adjustment are as described previously. The HbA1c value at the intermediary evaluation may also be used to guide treatment.

## End of intervention

Following final evaluations, intervention team members each meet with the participant to discuss strategies to maintain weight loss and exercise patterns. The study physician makes recommendations for medication management to the participant's usual treating physician, in conformity with national diabetes management guidelines and the participant's medication reimbursement plan regulations.

## Control group

The control group receives care from their usual physician, in accordance with National Institute for Health and Care Excellence[53] and Diabetes Canada[54] guidelines. Following final evaluations, control arm participants will be offered a 12-week low energy diet intervention followed by a 4-week food reintroduction period.

## Monitoring for adverse events

Intervention team members enquire about adverse events at each visit. Participants are asked to contact trial staff between visits to report any concerns. Serious adverse events will be reported within 7 days to all research ethics boards and the data monitoring committee, which may seek a formal adjudication of relatedness by physicians blinded to the treatment arm and who have not interacted with the study participants in question.

More common adverse events in low energy diet trials[9 55] include constipation (18%–47%), headache (8%), dizziness (4%–32%), fatigue (11%–25%) and thirst (6%). Constipation responds to fibre-containing laxatives and fluid intake, and other symptoms resolve over time. In one high intensity aerobic training trial,[56] 16% experienced fracture or lower extremity muscle cramping and/or muscle, ligament, or tendon strain; 5.2% experienced chest pain, difficulty breathing, dizziness or loss of consciousness. In a trial with a combined resistance and aerobic exercise arm,[19] 12.5% had one of shoulder injury, left knee pain, spinal stenosis exacerbation or hip pain; one participant experienced atrial fibrillation.

## Withdrawal

Interventions and evaluations are halted in the event of intolerance to the study intervention or an adverse event that precludes participation, prolonged or serious hospital admission, the development of a serious comorbid condition such as active malignancy or pregnancy. We will document reasons for participant-initiated withdrawal, if specified.

## COVID-19 restrictions and adaptations

If COVID-19 restrictions are resumed, exercise physiologist led sessions using video conferencing may be used to replace some in-person supervised exercise sessions, using heart rate monitors, self-monitoring devices and resistance bands. Any remaining in-person intervention visits or outcome measurement sessions will be undertaken using personal protective equipment following clinical guidelines that are current at the time of practice.

## Sample size

In the DIRECT trial, remission was 46% at 1 year (vs 4% in control arm; OR 19.7).[9] Even higher remission was observed over the shorter term in other studies.[57] We initially allowed for a minimum 45% remission at 24 weeks in the intervention group and up to a 5% remission rate in the control group. Using a Fisher's exact test with 90% power, a significance of 0.05, and a 1:1 ratio between intervention and control groups, we estimated needing 56 individuals to complete the trial (control=28, intervention n=28). Allowing for up to 30% drop-out, we planned to recruit 80 individuals in total (40 UK, 40 Canada) in our successful application for funding to the MRC and the Canadian Institutes of Health Research.

We subsequently obtained additional funding from the Canadian JR McConnell Foundation, and increased recruitment goals in Canada from 40 to 60 participants (40 UK, 60 Canada). With an overall sample size of 100 (40 UK, 60 Canada), if 70 (70%) complete the trial, we will have 90% power to detect a minimum 35% remission at 24 weeks in the intervention group and up to a 5% remission rate in the control group.

Retention of at least 35 participants in each arm (70 total) will provide over 80% power to detect a difference in mean peak early diastolic strain rate of 0.10/s difference, equivalent to the exercise effect in the DIASTOLIC trial (see BACKGROUND),[18] assuming that the common SD is 0.144 as in the DIASTOLIC trial and using a two group t-test with a 5% two-sided significance level.

## Statistical analyses

Our primary outcome measures the proportion of remission in the intervention arm relative to the control arm. We will apply intention-to-treat principles for the primary outcome, assuming that remission of diabetes did not occur if information is not available. Based on the DiRECT and DIADEM-I trials, we estimate that remission counts in the control arm could be less than 2. We will therefore use stratified Fisher's exact test statistics (ie, stratification by country and by sex) if remission counts are low in the control arm but consider Pearson $\chi^2$ test, if appropriate. If numbers permit, logistic ORs with 95% CIs (exact method) will also be performed to further quantify the efficacy of intervention over the control group, after adjusting for factors used to stratify the randomisation (country, age or sex); we will not have missing values for these covariates. Subgroup analyses for the primary outcome will be performed by country, sex and the degree of weight loss (<5%, 5%–10%, 10%–15%, ≥15% as examined in the DiRECT trial,[9] collapsing categories as needed, and by exercise adherence (<50%, >50% of supervised sessions attended). In a per-protocol analysis, we will restrict the intervention group to those who achieved at least 10% wt loss and adhered to at least two thirds of prescribed exercise sessions.

We will analyse dichotomous secondary outcomes (eg, diabetes remission at 12 weeks, hypertension remission at 12 and 24 weeks) as described for the primary outcome.

We will analyse continuous key secondary outcomes (eg, left ventricular peak early diastolic strain rate, lean soft tissue mass) through linear regression or log-linear regression, as appropriate, with stratification by country and sex.

## Process evaluation analyses

At least two trained team members will independently code transcripts for themes,[58] continually refining existing codes and identifying new ones. We will use Dedoose V.7.0.23 and NVivo analysis software to facilitate data coding/organisation.

## Data management

Data management is facilitated by institutional REDCap systems, with access controlled through Active Directory Technology. Data and images shared across participating centres use secure data transfer systems with all identifiable information removed. Within REDCap, we have derived plausible ranges for all outcome measures; values outside these ranges are automatically flagged and verified for accuracy.

## DISCUSSION

Low energy diets incorporating meal replacements can achieve T2DM remission through weight loss,[9–11] but may reduce lean mass.[18 19] Exercise can preserve lean mass[18 19] and directly lower insulin resistance.[13 20 50 59 60] Combined aerobic and resistance training yields additive glucose-lowering benefits.[20] Regular exercise reduces CVD events[61–64] and improves MRI-based measures of heart health.[18] RESET for REMISSION will quantify remission of T2DM in young adults achieved through a combined low energy diet and supervised aerobic and resistance training intervention and will also ascertain changes in diastolic function, lean mass and other factors that may impact long term outcomes.

Since funding was awarded for RESET for REMISSION, the literature has continued to evolve, with publication of the DIADEM-I trial, which targeted remission in younger adults less than 50 years of age of West Asian or North African ancestry. DIADEM-I combined a low energy diet (800–820 kcal/day) with exercise prescription from a trained professional centred on walking activities and self-monitoring tools, with resistance exercises introduced as the intervention progressed. While this trial did not include supervised exercise training, the objective measurement of physical activity, or cardiac function and structure, it does provide robust evidence that a low energy diet combined with physical activity can be combined in younger adults with T2DM while leading to an equivalent level of remission as has been shown in older populations, such as DiRECT.

While young persons with T2DM stand to gain the most from remission,[24] their daily life demands may render recruitment and adherence challenging. We have endeavoured to mitigate this by integrating virtual communication tools for monitoring and follow-up where possible. Recruitment itself may be challenging in these younger individuals, as they remain fewer in number than middle-aged and older individuals with T2DM.[27] We are, therefore, collaborating with a wide network of practices and drawing on other diabetes research cohorts in which participants have provided permission to be contacted for trials.

The international diabetes remission panel underscored the importance of evaluating the impacts of non-glycaemic measures during remission.[3–6] RESET evaluates not only lipid profiles and MRI measures of pancreatic, hepatic and visceral fat but also sensitive MRI-based indicators of heart health for a comprehensive picture of intervention impacts. Equally importantly, we seek to understand impacts on quality of life, diabetes distress and mood.

The Canada-UK collaboration that underpins our trial will allow a wider sharing of knowledge and perspectives than in a single-country study, challenging all to learn from best practices in each jurisdiction. The DiRECT trial led to an ongoing 5000 person pilot by the National Health Service in the UK for prescribing meal replacements as part of routine T2DM management.[53] The visibility of the current trial and the Canada-UK partnership could be an important step towards considering implementing such an intervention with integration of exercise in Canadian healthcare jurisdictions. Indeed, a future trial may seek partners beyond these two countries, to move from a demonstration of efficacy to one of effectiveness across a wide range of settings, and a plan for implementation. This is a critical journey not only to meet the glycaemic definitions of diabetes remission but also to establish its potential myocardial and vascular effects.

## Sponsors

The study is sponsored by the University of Leicester in the UK (RGOsponsor@leicester.ac.uk) and the Research Institute of the McGill University Health Centre in Canada (gilbert.tordjman@muhc.mcgill.ca) who will ensure the study is conducted according to Good Clinical Practice and that all contractual, governance, ethics and regulatory processes are followed.

## Trial steering committee

This committee is responsible for the overall oversight of the trial and will review and approve protocol amendments, any substudy proposals, and will review recruitment rates, protocol adherence, retention, compliance, safety issues, planned analyses and reports, and will act on recommendations of the data monitoring committee. The trial steering committee includes an independent chair (Jason Gill, University of Glasgow), an independent clinician (Alice Cheng, University of Toronto), the Nominated Principal Investigators (Kaberi Dasgupta, McGill University and Thomas Yates, University of Leicester), the key site investigator in Edmonton (Normand Boulé, University of Alberta) and the trial statistician (Elham Rahme, McGill University). The trial steering committee's

operations are described in its charter (online supplemental appendix 2).

## Data monitoring committee

The data monitoring committee is responsible for the interests and safety of the participants and its main role will be to make advisory recommendations to the trial steering committee. It includes an independent clinician (Andrew Farmer, Nuffield Department of Primary Care Health Sciences, University of Oxford), an independent chair (Janusz Kaczorowski, University of Montreal) and independent statistician (Stephen Sharp, University of Cambridge). The data monitoring committee is independent from the sponsors and operates in conformity with a charter (online supplemental appendix 3).

## Patient and public involvement

This group will provide feedback throughout the trial on recruitment, evaluation, and retention strategies and alternatives. They will be involved in interpretation of findings and knowledge dissemination efforts, including manuscript coauthorship, presentations and interviews. The partners in Canada include Josette Spencer, Mark Marcinkiewicz and Sylvie Lauzon, who is Executive Director of Diabetes Quebec. Patricia Kearns is a patient partner trainer who assisted with training of the patient partners. Alastair Masters is a patient partner based in the UK.

## DISSEMINATION

We will publish the results of this trial in peer-reviewed journals and disseminate further in the scientific community through educational and conference presentations and social media targeting health professionals and the scientific community. We will share findings with the general public through press releases, media interviews and social media forums. The trial patient partners and participants will be invited to assist with knowledge sharing, including media interviews and op-eds.

**Author affiliations**
[1]Department of Medicine, McGill University and Centre for Outcomes Research and Evaluation, Research Institute of the McGill University Health Centre, Montreal, Quebec, Canada
[2]Faculty of Kinesiology, Sport, and Recreation, University of Alberta, Edmonton, Alberta, Canada
[3]Diabetes Research Centre, University of Leicester and NIHR Leicester Biomedical Research Centre, University of Leicester and University Hospitals of Leicester NHS Trust, Leicester, UK
[4]School of Human Nutrition, McGill University, Montreal, Quebec, Canada
[5]Leicester Diabetes Centre, University Hospitals of Leicester NHS Trust and NIHR Leicester Biomedical Research Centre, University of Leicester and University Hospitals of Leicester NHS Trust, Leicester, UK
[6]Centre for Outcomes Research and Evaluation, Research Institute of the McGill University Health Centre, Montreal, Quebec, Canada
[7]School of Public Health, Imperial College London, London, UK
[8]Division of Endocrinology & Metabolism, Department of Medicine, University of Alberta, Edmonton, Alberta, Canada
[9]Department of Cardiovascular Sciences, University of Leicester and NIHR Leicester Biomedical Research Centre, University of Leicester and University Hospitals of Leicester NHS Trust, Leicester, UK
[10]Courtois Cardiovascular Signature Centre, McGill University Health Centre and Departments of Medicine and Diagnostic Radiology, McGill University, Montreal, Quebec, Canada
[11]Diabetes Research Centre, University of Leicester and NIHR Applied Research Collaboration - East Midlands (ARC-EM), University of Leicester and University Hospitals of Leicester NHS Trust, Leicester, UK
[12]Department of Agricultural, Food and Nutritional Science, University of Alberta, Edmonton, Alberta, Canada
[13]Department of Medicine, McGill University, Montreal, Quebec, Canada
[14]Department of Biomedical Engineering, University of Alberta, Edmonton, Alberta, Canada

**Contributors** KD and TY are the principal investigators of the trial and led the design and writing of this protocol, with critical input from coinvestigators NB and JH, particularly related to the exercise intervention and fitness and strength evaluations; SC, ER, FA and CMP, experts in nutrition and body composition measures; GPM, MGF, RBT and AD, experts in MRI-based assessments of cardiac function and ectopic fat; DDC and MH, related to health coaching and process evaluation; MM, DC, JC and JR as related to trial logistics and procedures; MJD, MS and RY, with important input on medication management; EG and KK, for comments on recruitment and overall procedures; ER, as related to statistical considerations; and IF, with respect to the data management plan.

**Funding** The primary funders of this trial are the Medical Research Council (MR/T031816/1) of the UK and the Canadian Institutes of Health Research (UCD-170584; Institutes of Musculoskeletal Research and of Nutrition, Metabolism & Diabetes), organisations that have contributed equally to this endeavour. Additional funding in Canada was obtained from the John R McConnell Foundation to support additional enrolment in Canada, as previously described. Support for intervention delivery at the UK site is provided by the NIHR Leicester Biomedical Research Centre and Clinical Research Network. For the Canadian sites, some capillary glucose meters and strips were donated by Roche Diabetes Care (Canada) and continuous glucose monitoring supplies by Dexcom (San Diego, California, USA).

**Disclaimer** The sponsors and funders have no direct role in study design, data collection, management, analysis, interpretation of data, report writing or publications that arise from the trial.

**Competing interests** MGF is board member, shareholder, and consultant of Circle Cardiovascular Imaging. Through the University Hospitals of Leicester NHS Trust, coinvestigator GM has a research agreement with Circle Cardiovascular Imaging. We will be using software from this company to analyse the cardiac MRI images that we will obtain. KK has received honoraria from AstraZeneca, Boehringer Ingelheim, Eli Lilly, Janssen, Merck Sharp & Dohme, Novartis, Novo Nordisk, Sanofi, Takeda, Servier, and Pfizer and research support from AstraZeneca, Boehringer Ingelheim, Eli Lilly, Merck Sharp & Dohme, Novartis, Novo Nordisk, Sanofi, and Pfizer. MJD has acted as consultant, advisory board member and speaker for Novo Nordisk, Sanofi, Lilly and Boehringer Ingelheim, an advisory board member and speaker for AstraZeneca, an advisory board member for Janssen, Lexicon, Servier and Gilead Sciences and as a speaker for Napp Pharmaceuticals, Mitsubishi Tanabe Pharma and Takeda Pharmaceuticals International. She has received grants in support of investigator and investigator initiated trials from Novo Nordisk, Sanofi-Aventis, Lilly, Boehringer Ingelheim, Astrazeneca and Janssen. TY has received investigator initiated funding for obesity-related research from AstraZeneca.

**Patient and public involvement** Patients and/or the public were involved in the design, or conduct, or reporting, or dissemination plans of this research. Refer to the Methods section for further details.

**Patient consent for publication** Not applicable.

**Provenance and peer review** Not commissioned; externally peer reviewed.

**Author note** Dr Dasgupta and Dr Yates are the Co-Lead investigators and Co-corresponding authors. They contributed equally to this work.

**ORCID iDs**
Kaberi Dasgupta http://orcid.org/0000-0002-2447-3553
Stéphanie Chevalier http://orcid.org/0000-0002-8497-1570
Deborah Chan http://orcid.org/0000-0001-5006-1651
Edward Gregg http://orcid.org/0000-0003-2381-6822
Abhishek Dattani http://orcid.org/0000-0001-7405-6056
Elham Rahme http://orcid.org/0000-0002-4168-4993
Gerry P McCann http://orcid.org/0000-0002-5542-8448

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
