## [Reviewer comments · BMJ Open]

ARTICLE DETAILS

TITLE (PROVISIONAL)	Remission of Type 2 Diabetes and Improved Diastolic Function by Combining Structured Exercise with Meal Replacement and Food Reintroduction among young adults: The RESET FOR REMISSION Randomized Controlled Trial Protocol
AUTHORS	Dasgupta, Kaberi; Boulé, Normand; Henson, Joseph; Chevalier, Stéphanie; Redman, Emma; Chan, Deborah; McCarthy, Matthew; Champagne, Julia; Arsenyadis, Frank; Rees, Jordan; Da Costa, Deborah; Gregg, Edward; Yeung, Roseanne; Hadjiconstantinou, Michelle; Dattani, Abhishek; Friedrich, Matthias G.; Khunti, Kamlesh; Rahme, Elham; Fortier, Isabel; Prado, CM; Sherman, Mark; Thompson, Richard; Davies, Melanie; McCann, Gerry; Yates, Thomas

VERSION 1 – REVIEW

REVIEWER	Hidrus, Aizuddin Universiti Malaysia Sabah, Public Health Medicine
REVIEW RETURNED	28-May-2022

GENERAL COMMENTS	First of all, the content of the study presented here is sound, and worth publication where:  a. it is written and well-structured b. clear objectives and worthwhile issues to fill up the gap in knowledge However, some issues need to be highlighted and resolved before they can be accepted for publication. Background  -To explain more on the differences between the types of trials (good AHEAD, DIRECT, DIADEM-I, RESET). Suggest to provide full of each abbreviation the first time they used -To provide/add brief statistics on the prevalence or incidence of type 2 diabetes and its relation to a healthy diet and physical exercise, globally and locally (United Kingdom). Methods  -This study was a randomised controlled trial. The authors are suggested to report according to CONSORT (checklists, diagram, flow chart, etc.) Sample size  -Why use Fisher's exact test? Fisher exact is a test to assess the association between two categorical variables when the assumptions of the Pearson Chi-Square test are not met. -To explain more whether it was two means or two proportions used. -Brief explanation on DIASTOLIC trial?
---

	Statistical analyses -How was the “Diabetes remission” measured? Using the HbA1c results? If yes, then Fisher’s exact test is not appropriate. Authors are suggested to use Mixed Factorial ANOVA to compare the HbA1c mean between the two groups. -If the “Diabetes remission” only started as “Yes” and “No” which are treated as categorical data, then Fisher’s exact is acceptable, but needs to explain why not Pearson Chi-Square? -To explain more on generalised linear modelling for the secondary outcomes. -To explain more on the “indicator method” that is applied for handling the missing data. Results -There is no results provided for this paper/study? Discussion -To add more comparisons/discussions between RESET trial with other types of trials (look AHEAD, DIADEM-I, DIASTOLIC) from previous studies. Author only discussed between RESET and DiRECT trials in the current discussion chapter.
--	---

REVIEWER	Sharin, Tasfia Islamic University
REVIEW RETURNED	26-Jul-2022

GENERAL COMMENTS	First of all I am really honored to review this article titled "Remission of Type 2 Diabetes and Improved Diastolic Function by Combining Structured Exercise with Meal Replacement and Food Reintroduction among young adults: The RESET FOR REMISSION Randomized Controlled Trial" Good wishes for your future trial. I have following comments for the improvement of this paper,  1. The main title should be capitalized form, e.g. 'METHOD' 2. I think the of headings and subheading is too much which disrupts the flow of reading. 3. The sample size could be increased for generalizing the outcomes. That's all from me, hope for the best for your article.
--

VERSION 1 – AUTHOR RESPONSE

Reviewer: 1

Dr. Aizuddin Hidrus, Universiti Malaysia Sabah

First of all, the content of the study presented here is sound, and worth publication where:

- a. it is written and well-structured**
- b. clear objectives and worthwhile issues to fill up the gap in knowledge**

However, some issues need to be highlighted and resolved before they can be accepted for

publication.

Many thanks for these comments.

Background

-To explain more on the differences between the types of trials (good AHEAD, DiRECT, DIADEM-I, RESET). Suggest to provide full of each abbreviation the first time they used

Thank you for this helpful suggestion. We have expanded our discussion of these trials in the Introduction and included the full title of each.

To provide/add brief statistics on the prevalence or incidence of type 2 diabetes and its relation to a healthy diet and physical exercise, globally and locally (United Kingdom).

We have added the following at the beginning of the introductory paragraph (BACKGROUND section):

The global prevalence of diabetes over 9% and is expect to surpass 10% by 2030. More than 90% is type 2 diabetes (T2DM). Diabetes is diagnosed in over 8% of Canadians and 6% of persons in the UK.

Reference:

Saeedi P, Petersohn I, Salpea P, Malanda B, Karuranga S, Unwin N, Colagiuri S, Guariguata L, Motala AA, Ogurtsova K, Shaw JE, Bright D, Williams R; IDF Diabetes Atlas Committee. Global and regional diabetes prevalence estimates for 2019 and projections for 2030 and 2045: Results from the International Diabetes Federation Diabetes Atlas, 9th edition. *Diabetes Res Clin Pract.* 2019 Nov;157:107843.

Methods

This study was a randomised controlled trial. The authors are suggested to report according to CONSORT (checklists, diagram, flow chart, etc.)

We prepared this protocol in concordance with the CONSORT checklist.

Sample size

Why use Fisher's exact test? Fisher exact is a test to assess the association between two categorical variables when the assumptions of the Pearson Chi-Square test are not met.

Our primary outcome measures the proportion of remission in the intervention arm relative to the control arm. Based on the DIRECT and DIADEM-I trials, we expect low remission counts in the control arms, which may be less than five. In this case, the Pearson chi-square test may not be a good fit as it applies an approximation assuming the large sample, while Fisher's exact test runs an

exact procedure of small samples. To enhance clarity, we have updated our description of statistical analyses as indicated below (edits underlined):

Our primary outcome measures the proportion of remission in the intervention arm relative to the control arm. We will apply intention-to-treat principles for the primary outcome, assuming that remission of diabetes did not occur if information is not available. Based on the DiRECT and DIADEM-I trials, we estimate that remission counts in the control arm could be less than 5. We will therefore use stratified Fisher's exact test statistics (i.e., stratification by country and by sex) if remission counts are low in the control arm and Pearson chi-square test if numbers are higher.

To explain more whether it was two means or two proportions used.

We will compare proportions. Please see the response above.

Brief explanation on DIASTOLIC trial?

We describe DIASTOLIC in the 'BACKGROUND' section of the original submission (see response to first reviewer comment). We have added some further details in the revised statistical analyses. To clarify in the SAMPLE SIZE section, we have added the comment '(see BACKGROUND)' when alluding to the DIASTOLIC trial.

Statistical analyses

How was the "Diabetes remission" measured? Using the HbA1c results? If yes, then Fisher's exact test is not appropriate. Authors are suggested to use Mixed Factorial ANOVA to compare the HbA1c mean between the two groups.

To clarify, the outcome is a dichotomous as remission yes/no. The definition of remission requires both an A1C of less than 6.5% and no glucose-lowering medications in the prior 12 weeks. As noted above, to enhance clarity, we have revised the statistical analysis section as follows (edits underlined):

Our primary outcome measures the proportion of remission in the intervention arm relative to the control arm. We will apply intention-to-treat principles for the primary outcome, assuming that remission of diabetes did not occur if information is not available. Based on the DiRECT and DIADEM-I trials, we estimate that remission counts in the control arm could be less than 5. We will

therefore use stratified Fisher's exact test statistics (i.e., stratification by country and by sex) if remission counts are low in the control arm but consider Pearson chi-square test, if appropriate.

If the “Diabetes remission” only started as “Yes” and “No” which are treated as categorical data, then Fisher’s exact is acceptable, but needs to explain why not Pearson Chi-Square?

Please see our response to the previous comment.

To explain more on generalised linear modelling for the secondary outcomes.

When we referred to GLM, we meant logistic regression for dichotomous variables and linear or log linear, as appropriate, for continuous variables. We have now opted to revise our description of the analyses of secondary outcomes as follows:

We will analyse dichotomous secondary outcomes (e.g., diabetes remission at 12 weeks, hypertension remission at 12 and 24 weeks) as described for the primary outcome. We will analyse continuous key secondary outcomes (e.g., left ventricular peak early diastolic strain rate, lean soft tissue mass) through linear regression or log-linear regression, as appropriate, with stratification by country and sex; we may use multiple imputation to impute missing outcomes for those lost to follow-up.

To explain more on the “indicator method” that is applied for handling the missing data.

We stated in the original submission that

Missing baseline data will be handled using the indicator method, so cases are not dropped where follow-up data is present without baseline.

On further reflection, we decided to remove this from the manuscript and analytic approach. This is a randomized controlled trial and key baseline variables will not be missing.

Results

-There is no results provided for this paper/study?

This is a protocol paper. The trial is ongoing and thus there are no results at present.

Discussion

-To add more comparisons/discussions between RESET trial with other types of trials (look AHEAD, DIADEM-I, DIASTOLIC) from previous studies. Author only discussed between RESET and DiRECT trials in the current discussion chapter.

We appreciate this comment. As noted above, we have provided more information on each of these trials in the Introduction and added further details about DIADEM-I in the revised version of the Discussion.

Reviewer: 2

Tasfia Sharin, Islamic University

Comments to the Author:

First of all I am really honored to review this article titled "Remission of Type 2 Diabetes and Improved Diastolic Function by Combining Structured Exercise with Meal Replacement and Food Reintroduction among young adults: The RESET FOR REMISSION Randomized Controlled Trial" Good wishes for your future trial.

Many thanks for these comments and the reviewer's well wishes.

I have following comments for the improvement of this paper,

1. The main title should be capitalized form, e.g. 'METHOD'

We agree that METHODS and AIMS are main section headers and should be capitalized. We have made these modifications.

2. I think the of headings and subheading is too much which disrupts the flow of reading.

We apologize for this. We hope that capitalizing the main section headings (see response to above comment) will mitigate any disruption in flow. We believe, however, that the subheadings are important in the organization of the paper.

3. The sample size could be increased for generalizing the outcomes.

We appreciate this. However, the sample size is driven by the budget and is appropriate for our aims in this PROBE efficacy trial. It is our intention to conduct a larger and longer effectiveness trial, if the efficacy trial demonstrates promising results. As we state in the final paragraph of the paper: *Indeed, a future trial may seek partners beyond these two countries, to move from a demonstration of efficacy to one of effectiveness across a wide range of settings, and a plan for implementation.*